# VECHR: A Dataset for Explainable and Robust Classification of Vulnerability Type in the European Court of Human Rights

**Shanshan Xu[1], Leon Staufer[1,2], Santosh T.Y.S.S[1],**
**Oana Ichim[3], Corina Heri[4], Matthias Grabmair[1]**
[1]Technical University of Munich, Germany, [2]LMU Munich, Germany
[3]Graduate Institute of International and Development Studies, Switzerland
[4]Faculty of Law, University of Zürich, Switzerland
{shanshan.xu, leon.staufer, santosh.tokala, matthias.grabmair}@tum.de
oana.ichim@graduateinstitute.ch, corina.heri@rwi.uzh.ch

## Abstract

Recognizing vulnerability is crucial for understanding and implementing targeted support to empower individuals in need. This is especially important at the European Court of Human Rights (ECtHR), where the court adapts convention standards to meet actual individual needs and thus to ensure effective human rights protection. However, the concept of vulnerability remains elusive at the ECtHR and no prior NLP research has dealt with it. To enable future work in this area, we present VECHR, a novel expert-annotated multi-label dataset comprised of vulnerability type classification and explanation rationale. We benchmark the performance of state-of-the-art models on VECHR from both the prediction and explainability perspective. Our results demonstrate the challenging nature of the task with lower prediction performance and limited agreement between models and experts. We analyze the robustness of these models in dealing with out-of-domain (OOD) data and observe limited overall performance. Our dataset poses unique challenges offering a significant room for improvement regarding performance, explainability, and robustness.

## 1 Introduction

Vulnerability encompasses a state of susceptibility to harm, or exploitation, particularly among individuals or groups who face a higher likelihood of experiencing adverse outcomes due to various factors such as age, health, disability, or marginalized social position (Mackenzie et al., 2013; Fineman, 2016). While it is impossible to eliminate vulnerability, society has the capacity to mitigate its impact. The European Court of Human Rights (ECtHR) interprets the European Convention of Human Rights (ECHR) to address the specific contextual needs of individuals and provide effective protection. This is achieved through various means, such as displaying flexibility in admissibility issues, and shifting the burden of proof (Heri, 2021).

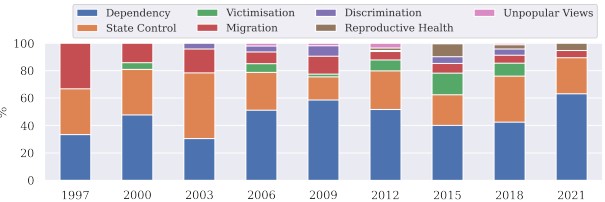

(a) Evolving distribution of types of vulnerability.

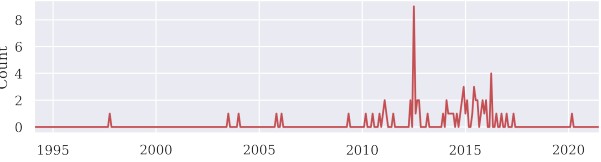

(b) Increase in application of the vulnerability type "migration" between 2010 and 2018.

Figure 1: Distribution changes of vulnerability types.

However, the concept of vulnerability remains elusive within the ECtHR. While legal scholars have explored vulnerability as a component of legal reasoning (Peroni and Timmer, 2013), empirical work in this area remains scarce and predominantly relies on laborious manual processes. To address this challenge, NLP can offer valuable tools to assist experts in efficiently classifying and analyzing textual data. Besides high classification performance, the true utility of NLP in the legal field is its ability to identify relevant aspects related to vulnerability in court cases. These aspects can be extracted, grouped into patterns, and used to inform both litigation strategy and legal policy. Even so, a significant obstacle to progress in this area is the lack of appropriate datasets. To bridge these research gaps, we present the dataset VECHR[1], which comprises cases dealing with allegation of Article 3 "Prohibition of torture" and is obtained from legal expert's empirical study[2]. Our proposed

---

[1]VECHR stands for **V**ulnerability Classification in **E**uropean **C**ourt of **H**uman **R**ights. Our dataset and code is available at https://github.com/TUMLegalTech/vechr_emnlp23

[2]Heri 2021. Heri is the fifth author of this work.

| Vulnerable Type | Description |
|---|---|
| Dependency | Including that of minors, the elderly, and those with physical, psychosocial and cognitive disabilities (i.e. mental illness and intellectual disability) |
| State Control | Including that of detainees, military conscripts, and persons in state institutions |
| Victimisation | Due to victimisation, including by domestic and sexual abuse, other violations, or because of a feeling of vulnerability |
| Migration | In the migration context, applies to detention and expulsion of asylum-seekers |
| Discrimination | Due to discrimination and marginalisation, which covers ethnic, political and religious minorities, LGBTQI people, and those living with HIV/AIDS |
| Reproductive Health | Due to pregnancy or situations of precarious reproductive health |
| Unpopular Views | Due to the espousal of unpopular views |
| Intersection | Intersecting vulnerabilities |

Table 1: Description of each vulnerability type. For more details, see App A.

task is to identify which type of vulnerability (if any) is involved in a given ECtHR case.

As model explainability is crucial for establishing trust, we extend the dataset with VECHR$_{explain}$, a token-level explanation dataset annotated by domain experts on a subset of VECHR. Its fine-grained token-level design mitigates performance overestimation of explainability when evaluated at the coarse paragraph level, as shown in previous works (Chalkidis et al., 2021; Santosh et al., 2022; Xu et al., 2023). Further, the understanding and application of vulnerability in court proceedings change over time, reflecting societal shifts and expanding to encompass a wider range of types (Fig 1a). The volume of cases also fluctuates significantly in response to social and political events (Fig 1b). To evaluate the model's robustness against distribution shifts, we further collect and annotate an additional out-of-domain (OOD) test set from cases involving non-Article 3 allegations, called VECHR$_{challenge}$.

We present comprehensive benchmark results using state-of-the-art (SOTA) models, revealing limited performance in vulnerability type classification in VECHR. We assess the models' alignment with expert explanations in VECHR$_{explain}$, and observe limited agreement. Experiment results on VECHR$_{challenge}$ indicate that, although incorporating description of the vulnerability type helps to improve the models' robustness, the performance remains low overall due to the challenges posed by the distribution shift. Our experiments underscore the difficulty of vulnerability classification in ECtHR, and highlight a need for further investigation on improve model accuracy, explainability, and robustness.

## 2 Vulnerability Typology in ECtHR

The inescapable and universal nature of vulnerability, as posited by Fineman (2016), underscores its significance in legal reasoning. For instance, the European Union has acknowledged the concept by establishing a definition for vulnerable individuals (Dir, 2013). However, it remains undefined within the context of ECtHR. To facilitate an examination of vulnerability and its application within the ECtHR, it is crucial to establish a typology recognized by the Court. Several scholars have endeavored to effectively categorize vulnerability for this purpose (Timmer, 2016; Limantė, 2022). One notable study is conducted by Heri (2021), which provides a systematic and comprehensive examination of the concept of vulnerability under ECHR Article 3. Heri proposes a complete typology encompassing eight types: *dependency*, *state control*, *victimization*, *migration*, *discrimination*, *reproductive health*, *unpopular views* and *intersections* thereof. Tab 1 gives a description for each type.

## 3 Data Collection and Annotations

### 3.1 Data Source and Collection Process

**VECHR** consists of 788 cases under Article 3, which were collected based on Heri's study of the Court's case law references of vulnerability. See App B for details on Heri's case sampling methodology and our post-processing procedures. We divided the dataset chronologically into three subsets: training (–05/2015, 590 cases), validation (05/2015–09/2016, 90 cases) and test (09/2016–02/2019, 108 cases).

**VECHR$_{explain}$**: We selected 40 cases (20 each) from the val and test splits for the explanation

dataset. Within each split, our sampling procedure involved two steps. First, we ensured coverage of all seven types by sampling one case for each type. Subsequently, we randomly selected an additional 13 cases to supplement the initial selection.

**VECHR_challenge**: To test the model's ability to generalize across distribution shifts, we extend VECHR by collecting and annotating additional cases *not* related to Article 3. Following Heri's method, we used the regular expression "vulne*" to retrieve all English relevant documents from the ECtHR's public database HUDOC[3] and exclude cases related to Article 3. We restricted the collection to the time span from 09/2016 (corresponding to start time of the test set) to 07/2022. In cases where multiple documents existed for a given case, we selected only the most recent document, resulting in a dataset consisting of 282 judgments. VECHR_challenge can be regarded as an out-of-domain topical (OOD) scenario. The in-domain train/val/test of VECHR are all from the same text topic cluster of Article 3. The OOD VECHR_challenge consists of non-Article 3 cases from different topic clusters (e.g. Article 10: freedom of expression), which involves different legal concepts and language usage.[4]

### 3.2 Vulnerability Type Annotation

We follow the typology and methodology presented by Heri 2021. She considered cases as "vulnerable-related", only when "vulnerability had effectively been employed by the Court in its reasoning". These cases are further coded according to the trait or situation (vulnerable type) giving rise to the vulnerability. In situations where the Court considered that multiple traits contributed to the vulnerability, she coded the case once for each relevant category. The resulting dataset comprises 7 labels[5]. Cases in which vulnerability was used only in its common definition, e.g. "financially vulnerability", were regarded as 'non-vulnerable' and were labelled none of the 7 types. See App C for more details of the definition of "vulnerable-related".

For cases under Article 3, we adopted the labelling provided by Heri's protocol. For

| Split | #C | T/C | P/C | #C_¬V | L/C | L/C_V |
|---|---|---|---|---|---|---|
| Train | 590 | 5140 | 83 | 325 | 0.70 | 1.55 |
| Validation | 90 | 5077 | 77 | 27 | 1.41 | 2.02 |
| Test | 108 | 3992 | 57 | 34 | 1.13 | 1.65 |
| Challenge | 282 | 4176 | 51 | 144 | 0.61 | 1.24 |
| Total | 1070 | 4765 | 72 | 530 | 0.78 | 1.54 |

Table 2: Dataset statistics for each split, with number of cases ($C$), number of non-vulnerable cases ($C_{\neg V}$), mean tokens ($T$) per case, mean paragraphs per case, mean labels ($L$) per case, and mean labels per case when only considering positive vulnerability cases ($C_V$).

VECHR_challenge, we ask two expert annotators[6] to label the case following Heri's methodology[7]. Each annotator has annotated 141 cases.

**Inter-Annotator Agreement** To ensure consistency with Heri's methodology, we conducted a two-round pilot study before proceeding with the annotation of the challenge set (details in App G). In each round, two annotators independently labelled 20 randomly selected cases under Article 3, and we compared their annotations with Heri's labels. The inter-annotator agreement was calculated using Fleiss Kappa, and we observed an increase from 0.39 in the first round to 0.64 in the second round, indicating substantial agreement across seven labels and three annotators.

### 3.3 Explanation Annotation Process

The explanation annotation process was done using the GLOSS annotation tool (Savelka and Ashley, 2018), see App H for details. Based on the case facts, the annotators was instructed to identify relevant text segments that indicate the involvement of a specific vulnerability type in the Court's reasoning. The annotators was permitted to highlight the same text span as an explanation for multiple vulnerable types.

### 4 Dataset Analysis

Tab 2 presents the key statistics of our dataset. VECHR comprises a total of 1,070 documents, with an average of 4,765 tokens per case ($\sigma = 4167$). 788 and 282 cases fall under the Article 3 and non-Article 3 partitions, respectively. Among all, 530 documents are considered as "non-vulnerable", meaning they are not labelled as any of the seven vulnerable types. In the vulnerable-

---

[3] https://hudoc.echr.coe.int

[4] For example, the Court recognizes the vulnerability of an elderly woman and provides her with protection under Article 3 (prohibiting torture) rather than Article 10 (freedom of expression)

[5] See App D for our justification for excluding the type "intersectionality".

[6] See App E for annotators' background and expertise.

[7] For the reason of why Heri confined her study to Article 3 and why the typology also applies to cases under other articles, please refer to App F.

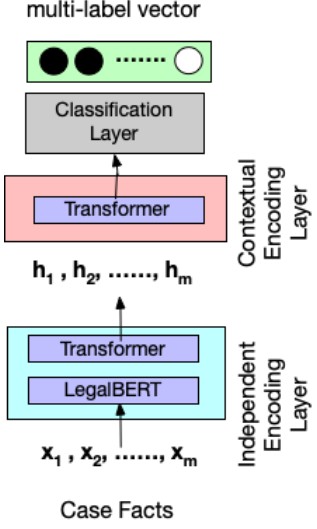

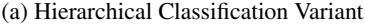

(a) Hierarchical Classification Variant

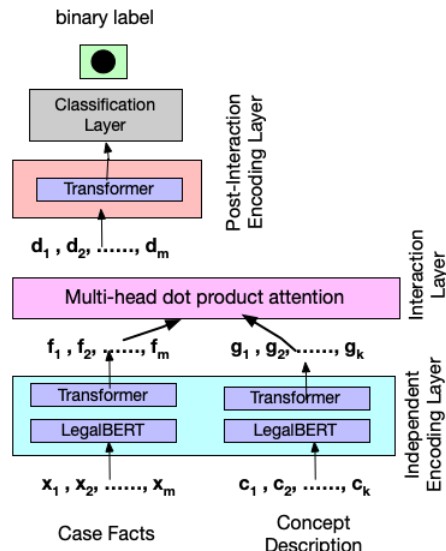

(b) Concept-aware Classification Variant

Figure 2: Visualization of Hierarchical and Concept-aware Hierarchical Model architectures.

related cases, the average number of labels assigned per document is 1.54.

We observe a strong label distribution imbalance within the dataset. The label "state control" dominates, accounting for 33% of the cases, while the least common label, "reproductive health", is present in only 3% of the cases. For more detailed statistics of our dataset, including details regarding the label imbalances in Tab 6, please refer to App I.

## 5 Experiments

### 5.1 Vulnerability Type Classification

**Task:** Our objective is to predict the set of specific vulnerability type(s) considered by the Court based on the factual text of a case.

**Models:** We finetune pre-trained models *BERT* (Devlin et al., 2019), *CaselawBERT* (Zheng et al., 2021), *LegalBERT* (Chalkidis et al., 2020): on our dataset with a multi-label classification head, truncating the input to the maximum of 512 tokens.

We finetune the *Longformer* model (Beltagy et al., 2020) on our dataset that allows for processing up to 4,096 tokens, using a sparse-attention mechanism which scales linearly, instead of quadratically.

We further employ a *hierarchical* variant of pre-trained LegalBERT to deal with the long input limitation. We use a greedy input packing strategy where we merge multiple paragraphs[8] into one packet until it reaches the maximum of 512 tokens.

[8]Details and statistics on paragraphs are reported in App I.

| Model | Classification | | Explanation |
|---|---|---|---|
| | mac-F1 | mic-F1 | Kappa |
| random | 19.02 | 25.07 | -0.11 ± 0.02 |
| BERT | 24.31 | 41.78 | 0.02 ± 0.06 |
| CaselawBERT | 27.31 | 45.16 | 0.04 ± 0.08 |
| LegalBERT | 27.34 | 42.47 | 0.04 ± 0.07 |
| Longformer | 31.49 | 46.21 | 0.11 ± 0.11 |
| Hierachical | 31.46 | 45.32 | 0.10 ± 0.08 |

Table 3: Classification and explanation results. We report F1s for classification performance and Kappa score with standard error for explanation agreement.

We independently encode each packet of the input text using the pretrained model and obtain representations ($h_{[CLS]}$) for each packet. Then we apply a non-pretrained transformer encoder to make the packet representations context-aware. Finally, we apply max-pooling on the context-aware packet representations to obtain the final representation of the case facts, which is then passed through a classification layer. Fig 2a illustrates the detailed architecture of the hierarchical model.

For details on all models' configuration and training, please refer to App J.

**Evaluation Metrics:** we report micro-F1 (mic-F1) and macro-F1 (macF1) scores for 7+1 labels, where 7 labels correspond to 7 vulnerability types under consideration and an additional augmented label during evaluation to indicate non-vulnerable.

**Results:** Tab 3 reports the results of classification performance. We observe that legal-specific pretraining improved the performance over general

| Model | VECHR$_{challenge}$ | |
|---|---|---|
| | mac-f1 | mic-f1 |
| random | 12.75 | 14.61 |
| BERT | 20.51 | 43.48 |
| CaselawBERT | 24.55 | 57.51 |
| LegalBERT | 22.60 | 50.77 |
| Longformer | 25.24 | 55.71 |
| Hierarchical | 26.43 | 58.46 |
| Concept-aware Hierarchical | **33.11** | 49.62 |

Table 4: Results on the challenge dataset.

pre-training. However, BERT models still face the input limitation constraint. Both Longformer and Hierarchical models improved compared to truncated variants and are comparable to each other. Overall, we see low overall performance across models, highlighting the challenging task.

### 5.2 Vulnerability Type Explanation

We use Integrated Gradient (IG) (Sundararajan et al., 2017) to obtain token-level importance from the model with respect to each vulnerable type under consideration. We max pool over sub-words to convert token-level IG scores into word-level scores, followed by a threshold-based binarization. Tab 3 reports explainability performance expressed as the average of Cohen's $\kappa$ between the models' focus and the experts' annotations for the test instances. We observe that the low explainability scores among different models reflect their trend in classification scores and also echo the challenging nature of the task.

### 5.3 Robustness to Distributional Shifts

We assess the robustness of models to distributional shift using the VECHR$_{challenge}$ and present the performance in Tab 4. Notably, we observe a drop in macro-F1 score on VECHR$_{challenge}$ compared to the test set. We attribute this to the models relying on suboptimal information about vulnerability types, which is primarily derived from the factual content rather than a true understanding of the underlying concept. To address this limitation, we propose a **Concept-aware Hierarchical** model that considers both the case facts and the description of vulnerability type to determine if the facts align with the specified vulnerability type[9], inspired by Tyss et al.

---

[9]We cast the multi-label task into a binary classification setup by pairing the text with each vulnerability type. These binary labels are transformed into a multi-label vector for performance evaluation, to produce a fair comparison to multi-label models on the same metric.

2023a. We employ a greedy packing strategy as described earlier and use a hierarchical model to obtain the context-aware packet representations for each packet in the facts and concept description separately. Subsequently, we apply scaled-dot-product cross attention between the packet vectors of the facts (as Query) and concepts (as Keys and Values), generating the concept-aware representation of the facts section packets. A transformer layer is used to capture the contextual information of the updated packet vectors. Then we obtain the concept-aware representation of the case facts via max pooling and pass it through a classification layer to obtain the binary label. Fig 2b illustrates the detailed architecture of the concept-aware model. For more details, see App K.

The concept-aware model exhibits increased robustness to distributional shift and shows an improvement on the challenge set, owed to the incorporation of the vulnerability type descriptions. Overall, our results show promise for the feasibility of the task yet indicate room for improvement.

## 6 Conclusion

We present VECHR, an ECtHR dataset consisting of 1,070 cases for vulnerability type classification and 40 cases for token-level explanation. We also release a set of baseline results, revealing the challenges of achieving accuracy, explainability, and robustness in vulnerability classification. We hope that VECHR and the associated tasks will provide a challenging and useful resource for Legal NLP researchers to advance research on the analysis of vulnerability within ECtHR jurisprudence, ultimately contributing to effective human rights protection.

### Limitations

In our task, the length and complexity of the legal text require annotators with a deep understanding of ECtHR jurisprudence to identify vulnerability types. As a result, acquiring a large amount of annotation through crowdsourcing is not feasible, leading to limited-sized datasets. Additionally, the high workload restricts us to collecting only one annotation per case. There is a growing body of work in mainstream NLP that highlights the presence of irreconcilable Human Label Variation(Plank, 2022; Basile et al., 2021) in subjective tasks, such as natural language inference (Pavlick and Kwiatkowski, 2019) and toxic language detection (Sap et al., 2022). Future work should address this limitation

and strive to incorporate multiple annotations to capture a more and potentially multi-faceted of the concept of vulnerability.

This limitation is particularly pronounced because of the self-referential wording of the ECtHR (Fikfak, 2021). As the court uses similar phrases in cases against the same respondent state or alleging the same violation, the model may learn that these are particularly relevant, even though this does not represent the legal reality. In this regard, it is questionable whether cases of the ECtHR can be considered "natural language". Moreover, the wording of case documents is likely to be influenced by the decision or judgement of the Court. This is because the documents are composed by court staff after the verdict. Awareness of the case's conclusion could potentially impact the way its facts are presented, leading to the removal of irrelevant information or the highlighting of facts that were discovered during an investigation and are pertinent to the result (Medvedeva et al.). Instead, one could base the analysis on the so-called "communicated cases", which are often published years before the case is judged. However, these come with their own limitations and only represent the facts as characterized by the applicant applicant and not the respondent state. There are also significantly fewer communicated cases than decisions and judgements.

One of the main challenges when working with corpora in the legal domain is their extensive length. To overcome this issue, we employ hierarchical models, which have a limitation in that tokens across long distances cannot directly interact with each other. The exploration of this limitation in hierarchical models is still relatively unexplored, although there are some preliminary studies available (e.g., see Chalkidis et al. 2022). Additionally, we choose to freeze the weights in the LegalBERT sentence encoder. This is intended to conserve computational resources and reduce the model's vulnerability to superficial cues.

## Ethics Statement

Ethical considerations are of particular importance because the dataset deals with vulnerability and thus with people in need of special protection. In general, particular attention needs to be paid to ethics in the legal context to ensure the values of equal treatment, justification and explanation of outcomes and freedom from bias are upheld (Surden, 2019).

The assessment of the ethical implications of the dataset is based on the Data Statements by Bender and Friedman (2018). Through this, we aim to establish transparency and a more profound understanding of limitations and biases. The curation is limited to the Article 3 documents in English. The speaker and annotator demographic are legally trained scholars, proficient in the English language. "Speaker" here refers to the authors of the case documents, which are staff of the Court, rather than applicants. We do not believe that the labelling of vulnerable applicants is harmful because it is done from a legally theoretical perspective, intending to support applicants. The underlying data is based exclusively on the publicly available datasets of ECtHR documents available on HUDOC[10]. The documents are not anonymized and contain the real names of the individuals involved. We do not consider the dataset to be harmful, given that the judgments are already publicly available.

We are conscious that, by adapting pre-trained encoders, our models inherit any biases they contain. The results we observed do not substantially relate to such encoded bias. Nonetheless, attention should be paid to how models on vulnerability are employed practically.

In light of the aforementioned limitations and the high stakes in a human rights court, we have evaluated the potential for misuse of the vulnerability classification models. Medvedeva et al. (2020) mention the possibility of reverse engineering the model to better prepare applications or defences. This approach is, however, only applicable in a fully automated system using a model with high accuracy towards an anticipated decision outcome. As this is not the case for the models presented, we assume the risk of circumventing legal reasoning to be low. On the contrary, we believe employing a high recall vulnerability model could aid applicants and strengthen their legal reasoning. In a scholarly setting focused on vulnerability research, we do not think the model can be used in a detrimental way. Our research group is strongly committed to research on legal NLP models as a means to derive insight from legal data for purposes of increasing transparency, accountability, and explainability of data-driven systems in the legal domain.

There was no significant environmental impact, as we performed no pre-training on large datasets. Computational resources were used for fine-tuning

---

[10]https://hudoc.echr.coe.int

and training the models, as well as assessing the dataset. Based on partial logging of computational hours and idle time, we estimate an upper bound for the carbon footprint of 30 kg $CO_2$ equivalents. This is an insignificant environmental impact.

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

# A  Vulnerability Typology and Descriptions

Here is the typology of vulnerability in ECtHR (Heri, 2021):

- **Dependency**: dependency-based vulnerability, which concerns minors, the elderly, and those with physical, psychosocial and cognitive disabilities (i.e., mental illness and intellectual disability).

- **State Control**: vulnerability due to state control, including vulnerabilities of detainees, military conscripts, and persons in state institutions.

- **Victimisation**: vulnerability due to victimisation, including by domestic and sexual abuse, other violations, or because of a feeling of vulnerability.

- **Migration**: vulnerability in the migration context, applies to detention and expulsion of asylum-seekers.

- **Discrimination**: vulnerability due due to discrimination and marginalisation, which covers ethnic, political and religious minorities, LGBTQI people, and those living with HIV/AIDS.

- **Reproductive Health**: vulnerability due to pregnancy or situations of precarious reproductive health.

- **Unpopular Views**: vulnerability due to the espousal of unpopular views.

- **Intersection**: intersecting vulnerabilities.

Following is a detailed description of each type:
**Dependency** Dependency-based vulnerability derives from the inner characteristics of the applicant and thus concerns minors, elderly people, as well as physical, psychosocial and cognitive disabilities (i.e., mental illness and intellectual disability). The Court has built special requirements around these categories to be fulfilled by States as part of their obligations under the Convention. Minors: The Court often refers to children as a paradigmatic example of vulnerable people and made use of the concept of vulnerability to require States to display particular diligence in cases imposing child protection given, on the one hand, their reduced ability and/or willingness of complaining of ill-treatment and, on the other hand, their susceptibility to be exposed to traumatic experiences/treatment. Elderly: In many ways, vulnerability due to advanced age is a continuation of the vulnerability of children. All humans experience dependency at the beginning of life, and many experience it near the end. Intellectual and Psychosocial Disabilities: Intellectual disability may render those affected dependent on others – be it the state or others who provide them with support. Having regard to their special needs in exercising legal capacity and going about their lives, the Court considered that such situations were likely to attract abuse. Persons living with intellectual disabilities experience difficulties in responding to, or even protesting against, violations of their rights. In addition, persons with severe cognitive disabilities may experience a legal power imbalance because they do not enjoy legal capacity.

**State Control** Vulnerability due to state control includes detainees, military conscripts, and persons in state institutions. This type of vulnerability includes persons in detention, but also those who are institutionalised or otherwise under the sole authority of the state. When people are deprived of their liberty, they are vulnerable because they depend on the authorities both to guarantee their safety and to provide them with access to essential resources like food, hygienic conditions, and health care. In addition, the state often controls the flow of information and access to proof. Hence, the Court automatically applies the presumption of state responsibility when harm comes to those deprived of their liberty.

**Victimisation** Vulnerability due to victimisation refers to situations in which harm is inflicted by someone else. This type of vulnerability applies to situations of domestic and sexual abuse, and other type of abuse. The Court has also found that a Convention violation may, in and of itself, render someone vulnerable. Crime victims who are particularly vulnerable, through the circumstances of the crime, and can benefit from special measures best suited to their situation. Sexual and domestic violence are expressions of power and control over the victim, and inflict particularly intense forms of trauma from a psychological standpoint. Failing to recognise the suffering of the victims of sexual and domestic violence or engaging in a stigmatising response – such as, for example, the perpetuation of so-called 'rape myths' – represents a secondary

victimisation or 'revictimisation' of the victims by the legal system

**Migration** Vulnerability in the context of migration applies to detention and expulsion of asylum-seekers. Applicants as asylum-seeker are considered particularly vulnerable based on the sole experience of migration and the trauma he or she was likely to have endured previously'. The feeling of arbitrariness and the feeling of inferiority and anxiety often associated with migration, as well as the profound effect conditions of detention in special centres, indubitably affect a person's dignity. The status of the applicants as asylum-seekers is considered to require special protection because of their underprivileged (and vulnerable) status.

**Discrimination** Vulnerability due to discrimination and marginalisation covers ethnic, political and religious minorities, LGBTQI people, and those living with HIV/AIDS. The Court recognises that the general situation of these groups – the usual conditions of their interaction with members of the majority or with the authorities – is particularly difficult and at odds with discriminatory attitudes. Similarly to dependency-based vulnerability, this type of vulnerability imposes special duties on states and depends not solely on the inner characteristics of applicants but also on their choices which, in most cases, states have to balance against other choices and interests.

**Reproductive Health** Vulnerability due to pregnancy or situations of precarious reproductive health concerns situations in which women may find themselves in particular vulnerable situations, even if the Court does not consider women vulnerable as such. This may be due to an experience of victimisation, for example in the form of gender-based violence, or due to various contexts that particularly affect women. Sometimes, depending on the circumstances, pregnancy may be reason enough for vulnerability while other times vulnerability is linked to the needs of the unborn children.

**Unpopular Views** Vulnerability due to the espousal of unpopular views includes: demonstrators, dissidents, and journalists exposed to ill-treatment by state actors. Where an extradition request shows that an applicant stands accused of religiously and politically motivated crimes, the Court considers this sufficient to demonstrate that the applicant is a member of a vulnerable group. Similarly to the case of victimisation, it is also the applicant's choice to display such views and vulnerability comes from particular measures that state undertake when regulating or disregarding such choices.

## B   More details on Data Source and Collection Process

Heri 2021 reported the details of the case sampling process. The following serves as a summary of her case sampling methodology: She used the regular expression "vulne*" to retrieve all relevant English documents related to Article 3 from HUDOC, the public database of the ECtHR, excluding communicated cases and legal summaries, for the time span between the inception of the Court and 28 February 2019. This yielded 1,147 results.

Heri recorded her labeling in an Excel sheet, including the application number for each case. The application number serves as a unique identifier for individual applications submitted to the ECtHR. To create VECHR, we fetch all relevant case documents from HUDOC, including their metadata. During the post-processing, when one case has multiple documents, we keep the latest document and discard the rest, which yields 788 documents.

## C   Definition of "Vulnerable-related"

Heri (2021) specified that only cases where "vulnerability had effectively been employed by the Court in its reasoning" are regarded as "vulnerable-related". Cases in which vulnerability was used only in its common definition, or used only in the context of other ECHR rights, or by third parties, were excluded. To ensure clarity and alignment with Heri's perspective, we communicated with her during the annotation process to clarify the definition of "vulnerable-related". As a result, we determined that vulnerability is labeled (primarily) in situations where:

- Vulnerability is part of the Court's main legal reasoning.
- The alleged victims (or their children) exhibit vulnerability.
- The ECtHR, rather than domestic courts or other parties, consider the alleged victims vulnerable.

## D   Omitting "Intersectionality" Label

The vulnerability type "intersectionality" was omitted because of its unclear usage in cases. Even

more than the other typologies, it is a highly nuanced concept without a clear legal definition. Furthermore, Heri (2021, 117) says that the ECtHR does not engage with the concept of intersectionality in this form. Given that there are only 11 cases exclusively annotated as "intersectional" (out of a total of 59), the effect of disregarding it in this work is negligible. Omission does not suggest that intersectionality fails to play a role in the reasoning of the ECtHR or that we deem it irrelevant. Instead, we suggest exploring the concept of vulnerability as a combination of the other seven vulnerabilities.

## E    Annotator Background & Expertise

Two annotators performed the classification task. Annotator 1 (the fourth author) is a Post-doctoral Researcher at a European Research Centre, who has worked at the European Court of Human Rights as a case lawyer. Annotator 2 (the second author) is a law student, with also a philosophy and computer science background. Prior to annotation, both annotators had read Heri's book, and Annotator 2 had received a training session on the ECHR from Annotator 1.

The explanation dataset was annotated by Annotator 1.

## F    Justification of Article Applicability

Heri's (2021) typology is limited to Article 3 of the ECHR, which pertains to the Prohibition of Torture. Under Article 3, the concept of vulnerability was first coined by the ECtHR, given that it deals with inhuman, degrading treatment and torture, which represent prototypical contexts of vulnerability. As such, an initial exploration under Article 3 is reasonable.

Applying Heri's procedure to non-Article 3 cases is nonetheless justified according to our legal expert because Heri's underlying typology is based on Timmer (2016) and relates to all articles. Furthermore, vulnerability is now a concept that is not limited to a single article, and which the ECtHR applies across articles.

## G    Pilot study for Annotating Non-Article 3 Cases

In the first round, both annotators independently labeled 20 randomly selected cases under Article 3. After completing the labeling process, they compared their annotations with Heri's labels and engaged in a discussion to address any discrepancies and clarify their understanding of the vulnerability concept. In the second round, the annotators independently labeled another 20 randomly selected cases. We calculated the inter-annotator agreement using Fleiss Kappa to measure the consistency between Heri's labels and the annotations provided by our two annotators. The Fleiss Kappa agreement increased from 0.39 in the first round to 0.64 in the second round, which we consider to be substantial agreement in a multi-label setting involving seven vulnerable types and three annotators.

## H    GLOSS Annotation Tool

The task of explanation annotation was done using the GLOSS annotation tool (Savelka and Ashley, 2018). Fig 7 demonstrates the GLOSS annotation interface.

| Case fact | |
| --- | --- |
| # cases | 40 |
| Avg. # vulnerable type per case | 1.3 |
| Avg. length per case | 2964 ±1991 words |
| Rationals from annotator 1 | |
| Avg. length case-allegation pair | 630 ± 551 words |

Table 5: Statistics for the explanation dataset.

| | Count | % | IRLbl |
| --- | --- | --- | --- |
| Dependency | 254 | 23.74 | 1.41 |
| State Control | 358 | 33.46 | 1.00 |
| Victimisation | 70 | 6.54 | 5.11 |
| Migration | 71 | 6.64 | 5.04 |
| Discrimination | 34 | 3.18 | 10.53 |
| Reproductive Health | 11 | 1.03 | 32.55 |
| Unpopular Views | 33 | 3.08 | 10.85 |
| Non-Vulnerable | 530 | 49.53 | 0.68 |

Table 6: Count, percentage of total documents (%), imbalance ratio per label (IRLbl) (Charte et al., 2013) for each type of vulnerability.

## I    Dataset Statistics

The dataset comprises 1070 cases. On average, each case involves 0.78 vulnerable types and 1.54 vulnerable types for non-negative cases. Fig 3 presents the distribution of the number of annotated labels per document. We report the imbalance characteristics for each label in Tab 6. Fig 4 shows the difference in frequency of vulnerability annotations between Article 3 and non-Article 3. Fig 5 illustrates the difference in frequency of each

vulnerability label between all four datasets. Tab 5 shows the statistics for the explanation dataset.

The hierarchical nature of the dataset is based on the naturally occurring paragraphs in the judgment texts. On average, each case consists of 71.54 paragraphs ($\sigma = 67.54$). The distribution of the number of paragraphs by vulnerability type is shown in Fig 6a. The mean token count is 4,765; its distribution by vulnerability type is depicted in Fig 6b. The distribution of the mean token count per paragraph by vulnerability type is shown in Fig 6c.

## J   Implementation Details

We use *BERT* "bert-base-uncased" (Devlin et al., 2019), *CaselawBERT* "casehold/legalbert" (Zheng et al., 2021), *LegalBERT* "nlpaueb/legal-bert-base-uncased" (Chalkidis et al., 2020), and *Longformer* "allenai/longformer-base-4096" (Beltagy et al., 2020). We finetune pre-trained models from the Transformers Hub (Wolf et al., 2020) on our dataset with a multi-label classification head, truncating to maximum lengths of 512 and 4096 tokens for BERT and Longformer models, respectively.

**Hyperparameter & Overfitting Measures**: For the BERT-based models, we perform a grid search for hyperparameters across the search space of batch size [4, 8, 16] and learning_rate [5e-6, 1e-5, 5e-5, 1e-4]. We train models with the Adam optimizer up to 8 epochs. We determine the best hyperparameters on the dev set and use early stopping based on the dev set macro-F1 score. For the Hierarchical models, we employ a maximum sentence length of 128 and document length (number of sentences) of 80. The dropout rate in all layers is 0.1. We perform a grid search for the hyperparameters across the search space of batch size [2, 4] and learning_rate [1e-6, 5e-6, 1e-5]. We train models with the Adam optimizer up to 20 epochs. We determine the best hyperparameters on the dev set and use early stopping based on the dev set macro-F1 score. We use *PyTorch* (Paszke et al., 2019) 2.0.1.

## K   Model Architecture

**Hierarchical Model:** Greedy packing turns the case facts input into $m$ packets as $x = \{x_1, x_2, \ldots, x_m\}$, where packet $x_i = \{x_{i1}, x_{i2}, \ldots, x_{in}\}$ consists of $n$ tokens. We pass each packet $x_i$ independently into the pre-trained LegalBERT model (Chalkidis et al., 2020) to extract the $h_i^{cls}$ representation for each packet.

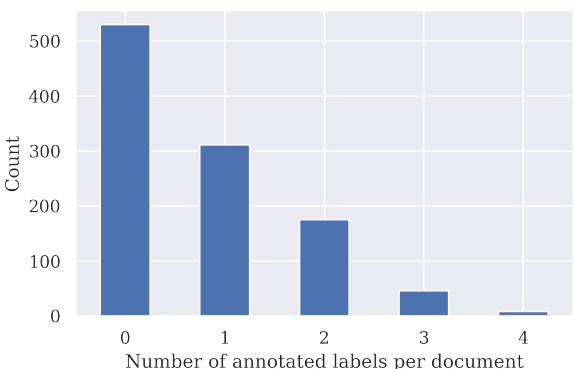

Figure 3: Distribution of number of annotated labels per document.

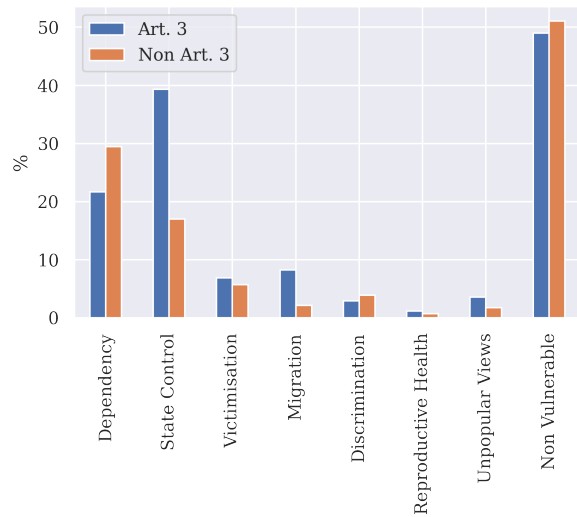

Figure 4: Difference in frequency of vulnerability annotations between Article 3 and non-Article 3.

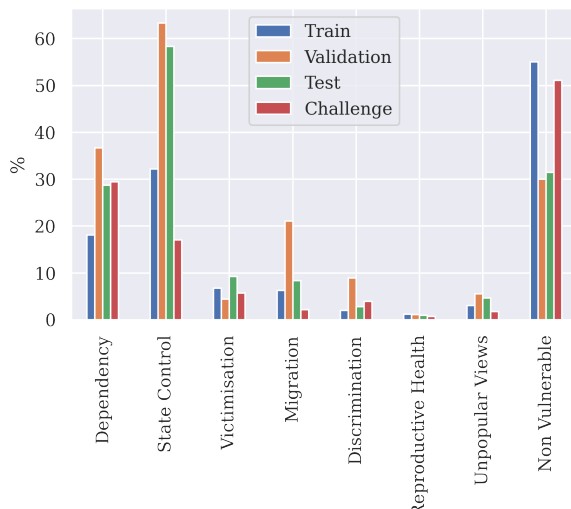

Figure 5: Difference in frequency of vulnerability type between "train", "validation", "test", and "challenge" datasets.

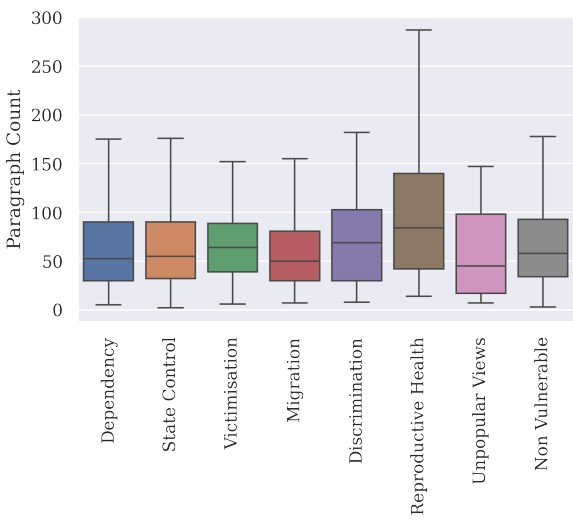

(a) Box plot of the number of paragraphs per type.

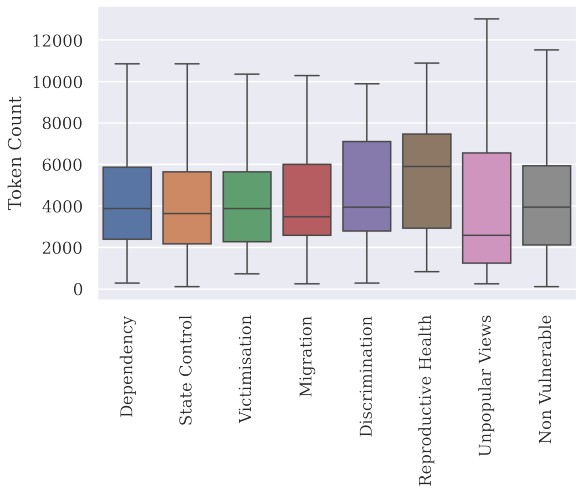

(b) Box plot of the token count per type.

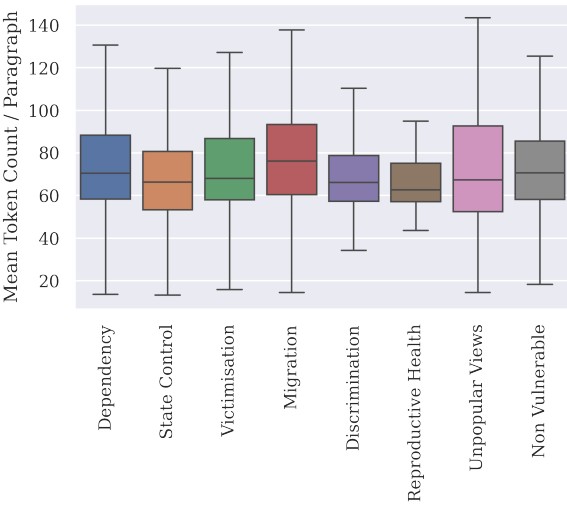

(c) Box plot of the mean number of tokens per paragraph per type.

Figure 6: Box plots on paragraph and token information for each vulnerability type.

All packet representations $h = \{h_1, h_2, \ldots h_m\}$, along with learnable position embeddings, are passed through a transformer encoder to make them aware of the surrounding context. These context-aware packets representations are then max pooled to obtain the final representation of the case facts, which then pass through a classification layer. Given the multi-label nature of the task, we employ a binary cross-entropy loss over each vulnerability type label. Fig 2a illustrates the detailed architecture of our hierarchical model, which is inspired by (Santosh et al., 2022; Tyss et al., 2023b).

**Concept-aware Hierarchical model:** We cast the multi-label task into a binary classification setup where we pair case facts with each vulnerability type to predict whether this vulnerability type was raised by court in the case from which the facts section stems. Note that we transform these binary labels into a multi-label vector for evaluation, to make a fair comparison with multi-label models. This is based on the article-aware outcome prediction setting in Tyss et al. 2023a.

The concept-aware model also takes the case facts as input, which after greedy packing form $m$ packets $x = \{x_1, x_2, \ldots, x_m\}$ and vulnerability concept description text $c = \{c_1, c_2, \ldots, c_k\}$ with $k$ packets. Packet $x_i = \{x_{i1}, x_{i2}, \ldots, x_{in}\}$ has $n$ tokens and packet $c_i = \{c_{i1}, c_{i2}, \ldots, c_{ip}\}$ has $p$ tokens.

Similar to the hierarchical model, we use a pretrained LegalBERT model (Chalkidis et al., 2020) to encode each packet in the case facts and concept description independently, and extract the $h^{cls}$ representation for each packet. These are passed through non-pretrained transformer model to obtain context-aware representations $f = \{f_1, f_2, \ldots, f_m\}$ and $g = \{g_1, g_2, \ldots g_k\}$ for case facts and concept descriptions, respectively.

The obtained packet representations of facts and concept description interact with each other using a multi-head scaled dot product cross attention (Vaswani et al., 2017) similar to the decoder in the transformer layer by treating case facts packets ($f$) as the queries (Q) and the keys (K) and values (V) come from the concept description packets ($g$).

$$\text{Attention}(Q, K, V) = \text{softmax}\left(\frac{QK^\top}{\sqrt{d_k}}\right)V \quad (1)$$

Thus we obtain a concept-aware representations of the fact description packets $d = \{d_1, d_2, \ldots, d_m\}$, which are once again passed through non-pretrained transformer module to enhance them with the surrounding contextual information, and max-pooled operation to obtain the final concept-aware case fact representation. A classification layer then predicts the binary label indicating whether these facts give rise to the given vulnerability type. Fig 2b displays the architecture of the concept-aware hierarchical model in detail.

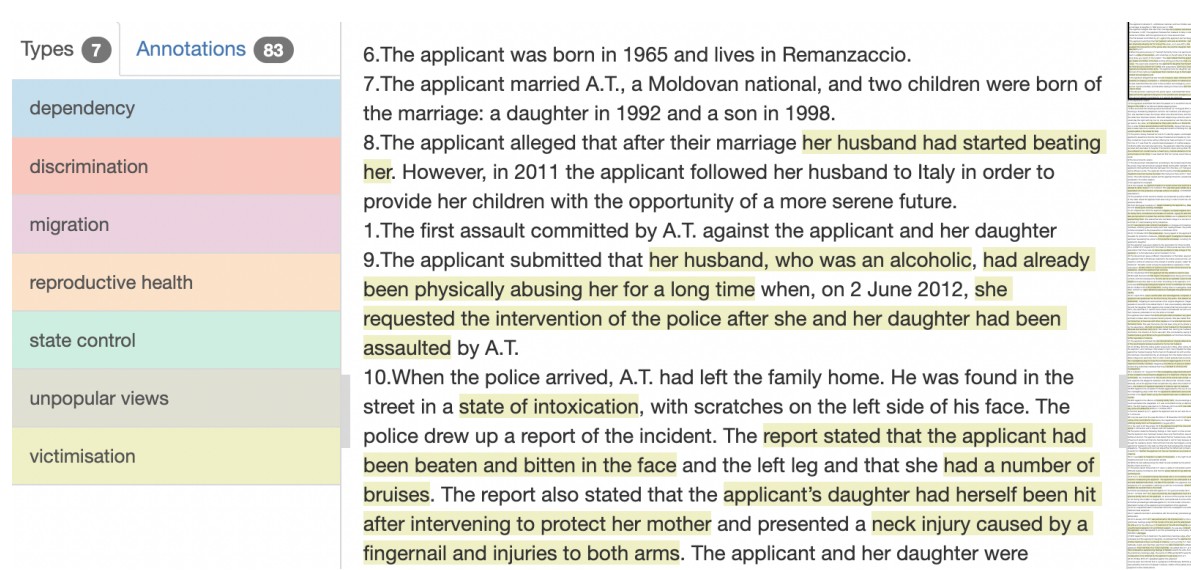

Figure 7: Screenshot of the GLOSS annotation interface.