# OpenReview forum: "VECHR: A Dataset for Explainable and Robust Classification of Vulnerability Type in the European Court of Human Rights"
_EMNLP/2023/Conference — EMNLP 2023 Main_

### Official Review · Reviewer_6aqx · 2023-07-27

**Soundness:** 4

**Excitement:**

4: Strong: This paper deepens the understanding of some phenomenon or lowers the barriers to an existing research direction.

**Missing References:**

Should cite papers related to packages used (pytorch, hugging face, etc.). We generally tend to forget their contribution and cite only models.

**Paper Topic And Main Contributions:**

The paper presents a novel expert annotated dataset for the important task of classifying vulnerability in the legal documents of the European Court of Human Rights.

It also benchmarks various language models by fine-tuning them on the proposed task.

It goes beyond creating a main dataset and creates two additional datasets to understand the explanation rationale and test out-of-distribution performance.


**Questions For The Authors:**

Q: Is the wrong base value used for calculating statistics reported on line #226? The sum is more than 100%. Please check and correct.

Q: Section 5.2 Table 1 suggestion: For explanation Kappa, as mean is reported, it will be useful to also report standard deviation.

Q: Why Concept-aware Hierarchical model is not used in experiments conducted in section 5.1?


**Reasons To Accept:**

R1: It constructs a dataset for an important task.
R2: The dataset is constructed very well, with details provided on annotation agreement, annotators’ backgrounds, etc.
R3: The paper is very well written. I learned some things about how to write a good appendix for a data resource paper. Even though many pages of the appendix are used, it is written very cleverly. Readers who are not reviewers or want to reproduce don’t need to read the appendix. This is particularly hard to do for a short paper. Kudos for that.
R4: Good well-thought effort is put into writing the limitations and ethics statement section.


**Reasons To Reject:**

It would be sad to reject such a good paper for just this reason but I will state it so they can improve it. Hyperparameter information for fine-tuning like batch size, and learning rate is missing. Also, what version of BERT is used is missing. It is better to specify a base or large, cased or uncased, and other details for each model.

**Reproducibility:**

4: Could mostly reproduce the results, but there may be some variation because of sample variance or minor variations in their interpretation of the protocol or method.

**Reviewer Confidence:**

4: Quite sure. I tried to check the important points carefully. It's unlikely, though conceivable, that I missed something that should affect my ratings.

**Typos Grammar Style And Presentation Improvements:**

Article 3 is defined first time on line #157 but was mentioned earlier on line #061. It will be useful to define it earlier.

I think there is a minor typo in footnote #3. Article numbers should be the opposite.

Lines #288-289: Word significant can be removed as a drop on micro is not much.

If the paper is accepted add details from Appendix J to the main paper utilizing the 5th page available.

Line #322: It should be 1,070 not 1070. A comma is missing for consistency.

Line #452: I agree with the statement, but it is not a good idea to make a judgment on what can have a significant or insignificant impact. As it doesn’t add any value, better to avoid writing it. More of a suggestion, please feel free to ignore it.

In Table 5: If I must guess the number followed by the +- is the standard deviation but it is better to explicitly state that.

I will suggest taking advantage of the additional 5th page (if accepted) to provide more discussion on results reported in section 5.

---

> ### Author Rebuttal · Authors · 2023-08-24
>
> Thanks for your detailed feedback.
>
> (4a) We will and should definitely add implementation details to the paper and also cite frameworks such as huggingface and pytorch.
>
> (4b) Line 226: Our task being Multilabel classification (i.e a datapoint can have multiple labels) and hence the sum can be over 100%
>
> (4c) Kappa-scores: We will add standard deviations.
>
> (4c) Concept-aware model has performance of 31.25 (mic-f1) and 43.11 (mac) on the test set, we would add that to Table 1.

---

### Official Review · Reviewer_DQPj · 2023-07-31

**Typos Grammar Style And Presentation Improvements:** Parenthese of Eq.1
**Soundness:** 3

**Excitement:**

3: Ambivalent: It has merits (e.g., it reports state-of-the-art results, the idea is nice), but there are key weaknesses (e.g., it describes incremental work), and it can significantly benefit from another round of revision. However, I won't object to accepting it if my co-reviewers champion it.

**Missing References:**

None

**Paper Topic And Main Contributions:**

The paper addresses the challenge of recognizing and classifying vulnerability types within the context of the European Court of Human Rights (ECtHR). The concept of vulnerability is vital in understanding individual needs and ensuring effective human rights protection. The author highlights that this concept remains elusive in the ECtHR, and there has been no prior NLP research addressing it.

Main Contributions:

1. **VECHR Dataset**: The authors present a novel expert-annotated multi-label dataset called VECHR, which includes vulnerability type classification and explanation rationale.
2. **Benchmarking Models**: The paper also benchmarks the performance of state-of-the-art models on the VECHR dataset from both prediction and explainability perspectives.

**Questions For The Authors:**

1. **Comparison with State-of-the-Art Methods**: How does the Concept-aware Hierarchical model compare with other state-of-the-art methods that utilize external knowledge? Could you provide a detailed analysis of why the mic-f1 score of the Concept-aware Hierarchical model is lower than that of the Hierarchical model as shown in Table 2?
2. **Handling Long Text Inputs**: Considering that the mean number of tokens in VECHR is 4596, while the maximum length of Longformer is only 4096, how did the authors handle instances that exceed this token limit? Have the authors considered applying retrieval or summarization methods to effectively reduce the input length and potentially improve the performance of Bert and Longformer?
3. **Hyperparameter Selection**: How were the hyperparameters, such as the learning rate and number of training epochs, selected for the experiments?
4. **Overfitting Concerns**: Considering that there are only 590 vulnerability cases in the training set, what measures were taken to prevent overfitting? Could you share the performance on the validation and test sets during the training process, along with any techniques used to mitigate overfitting?
5. **Inclusion of Large Language Models**: From my perspective, this dataset seems very suitable for testing on large language models such as GPT-4 and Claude-2, which can analyze each sentence and generate reasons belonging to specific types. Have the authors considered experimenting with these models?


**Reasons To Accept:**

1. **Novel Dataset Creation**: The introduction of the VECHR dataset is the main contribution. It fills a research gap by providing an expert-annotated multi-label dataset for vulnerability type classification, a previously unexplored area in NLP.
2. **Explainability Focus**: By including explanation rationale in the dataset, the paper aligns with the growing interest in explainable AI. This aspect allows for more interpretable models, crucial for applications in legal and social contexts.
3. **Interdisciplinary Relevance**: The paper's focus on human rights and legal applications showcases the breadth of NLP applicability, bridging the gap between technology and social sciences.


**Reasons To Reject:**

1. **Limited Generalizability**: The focus on a very specific legal context (ECtHR) may limit the generalizability of the findings and the applicability of the dataset to other legal systems or vulnerability assessments. A broader analysis or comparison with other contexts might have enhanced the paper's value.
2. **Distribution of content:** For a work on a dataset, specific definitions of data types, the data collection process, and detailed distribution of the dataset are crucial. However, these contents are placed in the appendix, and there is a lack of necessary analysis in the main body.

**Reproducibility:**

3: Could reproduce the results with some difficulty. The settings of parameters are underspecified or subjectively determined; the training/evaluation data are not widely available.

**Reviewer Confidence:**

3: Pretty sure, but there's a chance I missed something. Although I have a good feel for this area in general, I did not carefully check the paper's details, e.g., the math, experimental design, or novelty.

---

> ### Author Rebuttal · Authors · 2023-08-24
>
> Thanks for your valuable comments.
>
> (3a) Limited Generalizability: The concept of vulnerable/protected groups is a common topic in the civil/human rights context internationally and domestically (see, e.g. for example in the inter-american human rights setting:  https://www.ilo.org/wcmsp5/groups/public/---dgreports/---inst/documents/genericdocument/wcms_818106.pdf). Additionally, specialized legal concepts (and sub-concepts) of importance to certain areas of law are a common occurrence in legal systems. As such, the task presented here is representative of a task where expert legal researchers focus on a particular group of cases centering on a legal term that occurs sparsely. In such situations, discriminative models trained on the whole dataset may not be sensitive enough to be helpful.
>
> (3b)Distribution of content: We ask for understanding that given limited space of 4 pages, we use the appendices for the details of data collection process and dataset statistics. If our paper is accepted, we will utilise the 5th page to add more details to the main article.
>
> (3c) Concept-aware models: We limit our external knowledge to the definitions of vulnerable types using the concept-aware model. We leave using external sources of knowledge like convention document, legal commentaries, precedent cases etc for future work.
>
> (3d) Long inputs: We truncate any tokens exceedings the input length of the baseline longformer model. Our hierarchical model does not have this limitation. Other possible approaches (summarization, etc.) can be explored in future work.
>
> (3e) Hyperparameter & Overfitting measures: For the BERT-based models, we perform a grid search for the hyperparametes across the search space of batch size [4,8,16], learning_rate [5e-6, 1e−5, 5e−5, 1e-4]. We train models with the Adam optimizer up to 8 epochs. We determine the best hyperparameters on the dev set and use early stopping based on the dev set macro-F1 score.
> For the Hierarchical models, we employ a maximum sentence length of 128 and document length (number of sentences) of 80. The dropout rate in all layers is 0.1. We perform a grid search for the hyperparametes across the search space of batch size [2, 4], learning_rate [1e-6, 5e-6, 1e−5]. We train models with the Adam optimizer up to 20 epochs. We determine the best hyperparameters on the dev set and use early stopping based on the dev set macro-F1 score.
>
> (3f) Inclusion of LLM: We have experimented with ChatGPT and the dataset using different prompting strategies using a temperature of 0. We have found that, as expected, the prompt phrasing has significant influence on the performance, to the point producing counterintuitive metrics. The best model we obtained reaches a macro-F1 of ~.26 and a micro-F1 of ~.31, which is still substantially below our best models. Notably, this is with ChatGPT having possibly seen the cases during pre-training. The latter would be true in principle of GPT-4 and Claude-2, with the latter only having been available after the submission deadline. Given the variation produced by prompting and data contamination, we consider an exploration of the use of generative models for small-scale specialized legal concept identification a different research question that would warrant a separate paper.

---

### Official Review · Reviewer_tTjb · 2023-08-17

**Soundness:** 4

**Ethical Concerns:**

Yes

**Excitement:**

4: Strong: This paper deepens the understanding of some phenomenon or lowers the barriers to an existing research direction.

**Paper Topic And Main Contributions:**

This paper provides a dataset for classification of vulnerability type. This dataset consists of 1070 cases for prediction and 40 cases for explanation. This is a novel task, and this dataset can facilitate this new task.

**Reasons To Accept:**

This paper provides the dataset VECHR, which can bridge the NLP tools with experts in efficiently classifying and analyzing vulnerability, where vulnerability is important in ECtHR.

**Reasons To Reject:**

1. This dataset is small, especially the annotation scale of explanation rationale. Due to the small size of the evaluation data, the evaluation results may be unstable.
2. This paper lacks the performance of LLM (such as LLaMA 13B, chatGPT) on this dataset.

**Reproducibility:**

4: Could mostly reproduce the results, but there may be some variation because of sample variance or minor variations in their interpretation of the protocol or method.

**Reviewer Confidence:**

3: Pretty sure, but there's a chance I missed something. Although I have a good feel for this area in general, I did not carefully check the paper's details, e.g., the math, experimental design, or novelty.

---

> ### Author Rebuttal · Authors · 2023-08-24
>
> Thanks for your valuable comments.
>
> (2a) Small Dataset Size: We agree that a dataset with a large size and balanced label distribution would  have been better for training purpsoses. However, it's important to consider that in specialized domains such as the law— for instance our task involving the classification of specialized legal concepts within certain areas of law—the presence of limited data presents a unique challenge. This challenge would further captivate interesting directions involving few-shot, zero-shot, learning with limited data.
>
> (2b)  LLM: We have experimented with ChatGPT on our curated dataset with different prompting strategies and a temperature of 0. We have found that, as expected, the prompt phrasing has significant influence on the performance, to the point producing counterintuitive metrics. The best model we obtained reaches a macro-F1 of ~.26 and a micro-F1 of ~.31, which is still substantially below our best models. Notably, this is with ChatGPT having possibly seen the cases during its pre-training phase.

---

### Official Review · Reviewer_S2UF · 2023-08-22

**Typos Grammar Style And Presentation Improvements:** 1. Fig 1 label is absent.
**Soundness:** 4

**Excitement:**

4: Strong: This paper deepens the understanding of some phenomenon or lowers the barriers to an existing research direction.

**Missing References:**

N/A

**Paper Topic And Main Contributions:**

This paper proposes a new dataset with legal cases dealing with Article-3 allegations in the European Court of Human Rights expert-annotated with possibly multiple vulnerability types. Three different test sets are proposed, the first one is meant to evaluate models on the multi-label vulnerability classification task, the second one requires models to additionally extract textual spans as explanations for the identified vulnerability types, and the third one contains legal cases involving non-Article 3 allegations, thereby evaluating the robustness of models against distributional shifts/out of domain data. Several models are benchmarked against all three test sets for the respective tasks.
Overall low results highlight the challenging nature of the task as well as scope for further analysis and improvement.

**Questions For The Authors:**

1.  In Section 4 "Dataset Analysis", you mention that among 1070 documents, 519 documents are considered as “non-vulnerable”. However, in Table 3, total 551 case documents are labelled as “non-vulnerable”. Similarly, in Section 4, the average number of labels assigned per document is mentioned to be 1.51, whereas in Table 3, it is 1.59. Could you please explain this discrepancy?
2. What do you mean by "positive vulnerability cases" in Table 3? Since the stats are different for columns L/C and L/CV, are we still comparing the same no. of legal case documents? In other words, are positive vulnerability cases less than total cases/documents considered? If yes, how many documents are finally included in the dataset?
3. In Section 4, you mention that 519 documents are considered as “non-vulnerable” among 1070 documents. Doesn't this make the dataset skewed towards the "non-vulnerable" class? Also, among the "vulnerable" documents, the majority belong to the "Dependency" and "State Control" classes. Have you taken any steps to address this skewness while training models. Since, different vulnerability types have different definitions, can you expect models trained on VECHR to generalize well given the imbalance in label distribution?
4. In VECHR, the train set contains case documents prior to 05/2015, whereas val and test sets contain documents from non-overlapping periods different from that of train set. From Fig. 1 on page 1, we see that the distribution of vulnerability types evolves over time from 2015 to 2021. That makes the inherent label distribution different in the train set when compared to the val or test sets. Do you think this could be the reason for poor performance of models?
5. The micro-f1 scores of the trained models on the VECHR (challenge) dataset are almost always better (except the random model) than the corresponding scores on the VECHR dataset. Could you please explain this observation?
6. Is the label imbalance distribution reported in Table 4 for the overall VECHR dataset? Also, I cannot correlate Table 4 with Fig. 3. For example, "state control" has 33.5% representation as per Table 4, whereas, as per Fig. 3, it is less than 30% for Article-3 cases and only around 15% for Non-Article-3 cases. How can the value reach upto 33%? Could you please report the label distribution of the test sets separately for VECHR and VECHR (challenge)?
7. Model Description given in appendix J is not very clear. Could you please include a model diagram to explain the steps?
8. In Section 3.2, could you please clarify whether you recruited two or three annotators? Compare lines 180 and 194. Also, what is the inter-annotator agreement on the VECHR test set (not the challenge set which is reported)?

**Reasons To Accept:**

1. First of all, the paper is very well motivated, presented and written. It was worth a read.
2. It is true that datasets are scarce in the legal domain. Given the challenges and intricacies of the domain, NLP models are still far away from producing trustworthy explainable results which could be directly consumed without human intervention. In this regard, the proposed dataset is a valuable contribution. It is additionally novel given the task of vulnerability type analysis from European legal cases.
3. The paper proposes multiple test sets to evaluate various facets of compared models. In this regard, the thought process of the authors and their efforts towards getting multiple test sets annotated is noteworthy. It adds value to the paper.
4. Data collection and annotation steps are thoroughly explained with appropriate details. Although I found certain discrepancies. Please refer to my questions below.
5. Despite being a short paper, the authors have made sure to include necessary details in the appendix as well.
6. While more recent large language models could have been tried out, given that this is primarily a dataset and benchmarking paper, I am satisfied with the experiments.
7. All compared models have fairly low scores, which gives ample scope to future researchers to come up with better NLP models over time.
8. Finally, limitations of the work are clearly mentioned.

**Reasons To Reject:**

Although I am inclined towards accepting the paper, I have a few concerns:
1. Some of the dataset statistics reported in Section 4 and appendix I do not seem to correlate.
2. The dataset size is small with close to 50% of the documents not labelled with any vulnerability label. Additionally, only 2 labels dominate the overall label distribution. This raises questions on the dataset quality as well the scope for future models to perform well on this dataset.
3. At least one generative model could have been tried out leveraging the descriptions of vulnerability types to improve the classification performance.
4. Description of the proposed "Concept-aware Hierarchical" model is not clear.

**Reproducibility:**

3: Could reproduce the results with some difficulty. The settings of parameters are underspecified or subjectively determined; the training/evaluation data are not widely available.

**Reviewer Confidence:**

4: Quite sure. I tried to check the important points carefully. It's unlikely, though conceivable, that I missed something that should affect my ratings.

---

> ### Author Rebuttal · Authors · 2023-08-24
>
> Thank you for your valuable comments.
>
> (1a) Discrepancy in the Dataset Analysis (question 1, 2, 6; rejection reason 1): Thanks for pointing it out. We will correct it to maintain consistency at both places. The final dataset consists of 1070 documents. We will also provide more details, for example the label distribution of the separately for VECHR test and VECHR challenge etc. We will also adjust the table 3 to make it less confusing.
>
> (1b) Limited Dataset Size and Quality Concern (rejection reason 2): We agree that a dataset with a large size and balanced label distribution would have been better for training purposes. However, it's important to consider that in specialized domains such as the law— for instance, our task involving the classification of specialized legal concepts within certain areas of law—the presence of limited data presents a unique challenge. This challenge would further captivate interesting directions involving few-shot, zero-shot, learning with limited data.
>
> Specifically, to maintain the quality of our curated dataset, we engaged with experts from the beginning in several rounds of discussions, conducting a pilot study with them and finally an adjudication process to arrive at the annotation guidelines, which is reflected in the higher IAA score.
>
> (1c) Dataset Skewness and Generalizability (question 3):  We agree VECHR is very label imbalanced; therefore we propose a Concept-aware Hierarchical model to deal with the skewness, in the hope that the model will consider both the case facts and the description of vulnerability type to determine vulnerability type, rather than just learning the statistical distribution of the vulnerability type. Regarding generalizability,  the task presented here is representative of a task where expert legal researchers focus on a particular group of cases centering on a legal term that occurs sparsely. In such situations, discriminative models trained on the whole dataset may not be sensitive enough to be helpful.
>
>
> (1d)  LLM (rejection reason 3): We have experimented with ChatGPT on our curated dataset with different prompting strategies and a temperature of 0. We have found that, as expected, the prompt phrasing has significant influence on the performance, to the point producing counterintuitive metrics. The best model we obtained reaches a macro-F1 of ~.26 and a micro-F1 of ~.31, which is still substantially below our best models. Notably, this is with ChatGPT having possibly seen the cases during its pre-training phase.
>
> (1e) Diagram for Concept-aware Hierarchical model (question 7; rejection reason 4):  Thank you for the suggestion. We will add a diagram of the model structure and make the description more clear.
>
> (1f) Better Micro-f1 scores on the challenge set (question 3, 5):  This outcome can be attributed to the label distribution within the challenge set, which exhibits a more pronounced skew towards the majority class compared to the test set. The observed discrepancy—improved micro F1 but decreased macro F1 on the challenge set—suggests potential difficulties in the models' adaptability to distributional shift. These findings underscore the need to fortify the models' robustness.
>
> (1g) Number of annotators (question 8): For the 788 cases under article 3 (VECHR train/val/test set), the initial labelling was obtained from Heri’s study on Court’s case law references of vulnerability (Heri 2021). We then have our two annotators who read this vulnerability typology from Heri and participated in a pilot study of 40 randomly chosen article 3 cases to arrive at the consensus (line 190). The reported IAA score was calculated for these 40 cases using both annotators and Heri’s labels.
>
> While for the 282 cases in the Challenge set, we asked two expert annotators; each of them annotated 141 cases.
>
>
> Corina Heri. 2021. Responsive Human Rights: Vulnerability, Ill-treatment and the ECtHR. Bloomsbury 548 Academic.

---

### Meta-Review · Area_Chair_gByp · 2023-09-17

**Recommendation:** 4

**Metareview:**

This paper introduces a dataset for research in legal NLP on assessing the "vulnerability type" (a technical legal term) of cases before the European Court of Human Rights.

Pros:
- Important task which proves to be challenging for today's NLP systems in preliminary experiments (sufficient for the scope of a short paper).
- Well-presented and well-written. It's obvious that a lot of careful thought has gone into this work, both in setting it up as well as in running the project, and then finally in the write-up. The paper was easy to follow and used a solid strategy for delegating details to the appendix, which can be consulted where details are sought, but that's certainly not necessary to understand the paper.
- Extensive data annotation efforts to ensure broad coverage of the data set, which are also thoroughly described.
- Focus on both NLP accuracy as well as explainability, which is critical in this domain.
- Compliments to the authors on well-considered Ethics and Limitations sections, critical in applications like this.

Cons:
- Limited generalizability as this is a fairly niche task (no less important, but still).
- Some implementation details (like hyperparameters) were missing in the review copy, but this will be addressed for the camera-ready, as per the review threads.

---

### Decision · Program_Chairs · 2023-10-07

**Decision:**

Accept-Main

**Comment:**

This paper introduces a dataset for research in legal NLP on assessing the "vulnerability type" (a technical legal term) of cases before the European Court of Human Rights.

Pros:
- Important task which proves to be challenging for today's NLP systems in preliminary experiments (sufficient for the scope of a short paper).
- Well-presented and well-written. It's obvious that a lot of careful thought has gone into this work, both in setting it up as well as in running the project, and then finally in the write-up. The paper was easy to follow and used a solid strategy for delegating details to the appendix, which can be consulted where details are sought, but that's certainly not necessary to understand the paper.
- Extensive data annotation efforts to ensure broad coverage of the data set, which are also thoroughly described.
- Focus on both NLP accuracy as well as explainability, which is critical in this domain.
- Compliments to the authors on well-considered Ethics and Limitations sections, critical in applications like this.

Cons:
- Limited generalizability as this is a fairly niche task (no less important, but still).
- Some implementation details (like hyperparameters) were missing in the review copy, but this will be addressed for the camera-ready, as per the review threads.